# Reliability and Discriminative Validity of Wearable Sensors for the Quantification of Upper Limb Movement Disorders in Individuals with Dyskinetic Cerebral Palsy

**DOI:** 10.3390/s23031574

**Published:** 2023-02-01

**Authors:** Inti Vanmechelen, Saranda Bekteshi, Helga Haberfehlner, Hilde Feys, Kaat Desloovere, Jean-Marie Aerts, Elegast Monbaliu

**Affiliations:** 1Research Group for Neurorehabilitation (eNRGy), Department of Rehabilitation Sciences, KU Leuven, 8200 Bruges, Belgium; 2Department of Rehabilitation Medicine, Amsterdam Movement Sciences, Amsterdam UMC, 1081 HZ Amsterdam, The Netherlands; 3Research Group for Neurorehabilitation (eNRGy), Department of Rehabilitation Sciences, KU Leuven, 3000 Leuven, Belgium; 4Research Group for Neurorehabilitation (eNRGy), Department of Rehabilitation Sciences, KU Leuven, 3212 Pellenberg, Belgium; 5Department of Biosystems, Measure, Model & Manage Bioresponses (M3-BIORES), Division of Animal and Human Health Engineering, KU Leuven, 3000 Leuven, Belgium

**Keywords:** dyskinetic cerebral palsy, inertial measurement unit, upper limb, dystonia, choreoathetosis

## Abstract

**Background**—Movement patterns in dyskinetic cerebral palsy (DCP) are characterized by abnormal postures and involuntary movements. Current evaluation tools in DCP are subjective and time-consuming. Sensors could yield objective information on pathological patterns in DCP, but their reliability has not yet been evaluated. The objectives of this study were to evaluate (i) reliability and (ii) discriminative ability of sensor parameters. **Methods**—Inertial measurement units were placed on the arm, forearm, and hand of individuals with and without DCP while performing reach-forward, reach-and-grasp-vertical, and reach-sideways tasks. Intra-class correlation coefficients (ICC) were calculated for reliability, and Mann–Whitney U-tests for between-group differences. **Results**—Twenty-two extremities of individuals with DCP (mean age 16.7 y) and twenty individuals without DCP (mean age 17.2 y) were evaluated. ICC values for all sensor parameters except jerk and sample entropy ranged from 0.50 to 0.98 during reach forwards/sideways and from 0.40 to 0.95 during reach-and-grasp vertical. Jerk and maximal acceleration/angular velocity were significantly higher for the DCP group in comparison with peers. **Conclusions**—This study was the first to assess the reliability of sensor parameters in individuals with DCP, reporting high between- and within-session reliability for the majority of the sensor parameters. These findings suggest that pathological movements of individuals with DCP can be reliably captured using a selection of sensor parameters.

## 1. Introduction

Individuals with dyskinetic cerebral palsy (DCP) present with involuntary movements and intermittent muscle contractions, interfering with activities of daily life [1]. DCP is the second-most prevalent form of cerebral palsy (CP), the latter the most prevalent neurological childhood disability [2,3]. The movement disorders in DCP are subdivided into dystonia and choreoathetosis. Dystonia is defined by abnormal postures, involuntary twisting, and repetitive movements, while choreoathetosis is characterized by hyperkinesia and muscle tone fluctuation [1]. DCP is caused by a brain lesion, often in the thalamus and basal ganglia region [4]. Brain lesions in these regions frequently lead to movement disorders, severely affecting everyday-life activities and hindering an independent lifestyle. Due to these movement disorders, the majority of patients with DCP face significant challenges on a daily basis to execute functional activities such as eating and dressing [5]. The severity of dystonia was significantly associated with manual abilities as assessed with the Manual Ability Classification Scale (MACS), emphasizing the disabling aspect of this movement disorder and its detrimental impact on functional tasks [6,7].

Upper limb function or the execution of functional tasks requires fine-tuned coordination of multiple upper limb joints, which is disturbed in patients with dystonia and choreoathetosis [8]. This leads to slower movement, as well as increased variability during outward reaching for the DCP group in comparison to typically developing (TD) peers [9,10,11].

The severity of dystonia and choreoathetosis in DCP is currently evaluated by clinical evaluation scales such as the Burke–Fahn–Marsden Dystonia Rating scale or the Dyskinesia Impairment Scale [12,13,14]. In such scales, patients are video recorded while performing specific actions or during multiple rest postures, and the clinician subsequently scores the severity of dystonia and/or choreoathetosis for each body region according to pre-defined definitions per body region [13,14]. However, these scales are based on consensus definitions which may differ depending on the scale’s developer and require ordinal ratings of video observations, and are therefore prone to subjectivity [12,15]. Additionally, this evaluation method is time-consuming for physicians as it requires both acquisition and scoring of the videos. In order to objectify measurements of upper limb movement, three-dimensional motion analysis (3DMA) has been increasingly used in both research and clinical practice in typically developing (TD) individuals and individuals with CP [16,17,18]. It offers a reliable position estimation of anatomical landmarks over time, allowing calculations of upper limb joint angles and, subsequently, the description of abnormal movement patterns over time. In addition, we have recently shown that a minimum of eight repetitions should be included to reflect the variable movement pattern in children and adolescents with DCP [19]. Although 3DMA offers great insights into abnormal movement patterns, it cannot assess parameters such as movement complexity or movement smoothness. These measures may be significant in patients with uncontrolled, jerky movements because they can possibly detect the reduction in involuntary movements after an intervention.

In this respect, inertial measurement units (IMUs) or wearable sensors can be a valuable tool to evaluate the characteristics of upper limb movements in individuals with DCP [1]. IMUs contain an accelerometer and a gyroscope, enabling them to measure linear acceleration and angular velocity of the segment they are placed on. Compared to 3DMA, wearable sensors offer higher practicality and ease of use because they are placed on the body and their output data are directly sent to a computer. Such a set-up could significantly reduce the cost of an objective upper limb motion analysis, thus increasing its applicability in clinical practice. Furthermore, the mobile aspect of body-worn sensors and IMUs ensures higher comfort for the patient, and facilitates long-term and frequently repeated assessment. This is particularly important for patients who are unable to travel due to a severe mobility impairment and can thereby be evaluated in a home or school environment.

We have recently provided an overview of sensor parameters to evaluate movement disorders in the upper limb in multiple pathologies [20]. The majority of studies using IMUs to evaluate movement disorders focused on Parkinson’s Disease, where the main objectives were to discriminate patient groups from healthy controls or to correlate sensor parameters with clinical scores [21]. In stroke, upper limb rehabilitation was quantified using wearable sensors and correlation with clinical scales [22,23]. The CP population was included in only four studies. Multiple studies reported significant differences between the paretic and non-paretic arm in children with unilateral CP [11,24,25,26], but only one recent proof-of-concept study on home-based sensor measurements in participants with DCP is available [25]. In all abovementioned studies, choice of sensor parameters was scarcely explained, resulting in a multitude of sensor parameters, whilst only three studies examined the reliability of their sensor parameters [27,28,29].

Therefore, the first objective of this study was to evaluate within- and between-session reliability of sensor parameters in individuals with DCP. The second objective was to evaluate the discriminative validity of these sensor parameters between children and adolescents with DCP and their TD peers.

## 2. Materials and Methods

### 2.1. Participants and Study Setting

Children and youths with DCP and TD individuals were evaluated in this cross-sectional study. Participants with DCP were recruited from multifunctional centers for individuals with motor disabilities in Flanders, and from University Hospitals Leuven. The inclusion criteria for the DCP group were: (1) a diagnosis of DCP by a pediatric neurologist; (2) age of between 5 and 25 years; and (3) classified as MACS level I-III [30]. The MACS has shown good psychometric properties in addressing manual abilities in children with DCP [30,31]. The exclusion criteria were: (1) a neurological disorder other than DCP; (2) botulinum-toxin A injections within six months before assessment; and (3) neurological or orthopedic surgery in the year prior to assessment. All participants and/or their parents provided a written consent form prior to participation in accordance with the Declaration of Helsinki. The study was approved by the Medical-Ethical Committee, S-number S62093 (Commissie Medische Ethiek KU Leuven). Data collection was performed in the C-MALL of University Hospital Pellenberg and in the HTM-lab of KU Leuven Bruges.

### 2.2. Study Procedures

Participants were asked to perform three upper limb tasks: reaching forward (RF), reaching sideways (RS), and reach-and-grasp vertical (RGV). These tasks were part of a pre-existing protocol for three-dimensional motion analysis of upper limb tasks [19]. Five IMUs (XSens MTw Awinda; XSens, Enschede, The Netherlands) were placed on the scapula, sternum, upper arm, forearm, and hand according to a standardized sensor placement protocol. XSens IMUs have range of 2000 degrees/seconds for angular velocity and 160 m/s^2^ for linear acceleration, and a static and dynamic accuracy of 0.5 and 0.75 degrees RMS, respectively. For the purposes of this study, only the upper arm, forearm, and hand IMU were used (Appendix A). RF, RS, and RGV were performed at shoulder height (acromion), with a reaching distance determined according to arm length (i.e., from acromion to caput metacarpal III) (Appendix A). All tasks were performed at a self-selected speed with the non-preferred arm or with both arms in participants with bilateral DCP who agreed to have both arms evaluated. The reference position was 90° flexion in hip and knees and the hands placed on the ipsilateral knee, which was indicated with an elastic band around the knee [17]. Every task was executed 3 times with 10 repetitions per task execution, leading to 30 repetitions for each task. Each participant was evaluated twice on the same day by the same assessors with a minimum of one hour and a maximum of two hours between assessments. Sensors were taken off between sessions. All trials were video recorded for later clinical analysis. IMUs were sampled at 100 Hz and synchronized with video cameras to detect start-and-stop moments.

### 2.3. Data Analysis

The sensor data of the functional tasks were loaded into MT Manager (MT Manager, version 4.6; Xsens 2016, The Netherlands) to check for errors or signal loss and subsequently loaded into Matlab for further analysis (MathWorks, Natick, MA, USA). The raw acceleration and angular velocity signals were obtained, and the gravity vector was extracted from the raw acceleration data. Subsequently, sensor signals of the RF, RGV, and RS tasks were segmented according to time-synchronized video images using two different approaches: (i) each movement cycle was segmented separately, where a movement cycle was defined from hand on ipsilateral knee to point of task achievement (PTA) [17]; or (ii) all movement cycles within one task execution were considered as the full-length signal. In the separate movement cycles, the first and last repetition were excluded from the data to avoid start-and-stop strategies, leading to 8 repetitions per task execution and a total of 24 (3 × 8) repetitions per task (RF/RGV/RS). For the full-length signal, all eight repetitions per task were included (illustrated in Figure 1). Sensor parameters were calculated from both approaches and evaluated for within- and between-session reliability to assess the effect of signal length on parameter calculation.

Acceleration is defined as the change in velocity over time, whereas angular velocity data are defined as the rate of angular change over time. Acceleration and angular velocity were filtered with a fourth-order low-pass Butterworth filter, with a cut-off frequency of 10 Hz after visual check of the frequency spectrum, confirming no frequencies higher than 10 Hz. The norm of acceleration and gyroscope measures was calculated to minimize the influence of sensor location on data variability; subsequently, several sensor parameters were calculated from the norm values.

### 2.4. Sensor Parameters

The most prevalent sensor parameters in CP and DCP according to a recent review were execution time, mean, standard deviation (SD), and root mean square (RMS) of acceleration and angular velocity, jerk, and normalized jerk index [20]. Based on this overview and the unique characteristics of dystonia and choreoathetosis, we included mean, maximum and RMS of acceleration, and angular velocity. Maximal values of acceleration and angular velocity provide a representation of involuntary movements, whereas root-mean-square values provide information on signal variability over time. Additionally, we calculated maximal jerk for both acceleration and angular velocity. Jerk measures are hypothesized to represent the jerky, involuntary movements of choreoathetosis. Jerk is the derivative of acceleration and the second derivative of angular velocity, obtained by the following formulae:(1)$$jerk\;(j)=\frac{da}{dt}\;with\;a=acceleration$$
(2)$$angular\;jerk\;\left(\zeta \right)=\frac{d²\omega }{d²t}\;with\;\omega =angular\;velocity$$

Finally, the sample entropy of acceleration and angular velocity was obtained. Sample entropy assesses the complexity of time-series data by measuring the degree of dependency of a given data point on previous data points [32] and was calculated via Lee [33] with the embedding dimension *n* = 2 and the tolerance level set to 0.2 × SD(signal).

### 2.5. Statistical Analyses

Within-session reliability—Intra-class correlation coefficient (ICC) values and associated confidence intervals were calculated: (i) between sensor parameters of eight randomly selected movement cycles of each task (as this number is sufficient to capture the variable movement pattern in DCP) [19]; and (ii) between the sensor parameters of the full length-signal of the eight repetitions of the RF1, RF2, and RF3 tasks (similar for RGV and RS). Within-session reliability of the parameters was assessed with the intra-class correlation coefficient ICCw(2,1) based on single data [34].

Between-session reliability—Between-session ICC values and associated confidence intervals were calculated by: (i) comparing the mean of eight random repetitions in the test with the mean of eight repetitions in the re-test session; and (ii) comparing the full-length signal of each task (e.g., RF1) in the test session with the same task in the re-test session. Between-session reliability was evaluated with ICCb(2,k) based on averaged data. Additionally, between-session standard error of measurement (SEM) of the sensor parameters was calculated according to Schwartz et al. [35] for the long signal to compare with the between-group differences (see below).

Values of ICC were interpreted as poor (<0.50), moderate (0.50–0.75), good (0.75–0.90), and excellent (>0.90) [36].

Between-group differences—Normality of the data was assessed with the Shapiro–Wilk test. As the data were not normally distributed, the Mann–Whitney U test was used to compare sensor parameters of the mean full-length signal for each task between individuals with and without DCP. The mean absolute difference between the TD group and the DCP group was subsequently compared with the between-session SEM to evaluate for which sensor parameters the absolute difference between groups exceeded the between-session SEM value. 

All statistics were calculated using IBM SPSS Statistics 25 (SPSS Inc., Chicago, IL, USA).

## 3. Results

### 3.1. Participants

This study included 22 extremities of 18 participants with DCP (mean age 16y8m, SD 5y6m, 15 boys) and 20 extremities of TD peers with the same age range (mean age 17y3m, SD 3y8m, 8 boys). Participant characteristics are summarized in Appendix A. In the TD group, the left arm was measured for 16 participants and the right arm for 4 participants. In the DCP group, the left arm was measured 12 times and the right arm 10 times. For between-session reliability, five participants with DCP did not have data for the retest session. Two individuals with DCP were measured a second time, but these data were not valid due to technical issues. One individual was not measured a second time because of fatigue, and two individuals were not measured a second time because of time issues. The between-session results are thus based on 17 instead of 22 extremities. For three participants with DCP, the hand was too small for the hand IMU. The results from the hand sensor are thus based on 19 extremities for within-session reliability and 14 extremities for between-session reliability. 

### 3.2. Reliability

Within-session and between-session ICC values for the DCP group for the full tasks and the separate repetitions are reported in Figure 2 and Figure 3 for all sensors and all tasks. Additionally, ICC values and confidence intervals (CIs) are reported in Appendix A.

Within-session reliability: Reach forward—All ICC values for the hand sensor were higher for the full-task signal in comparison with the separate repetitions. For the full-task signal, ICC values ranged from moderate to excellent (0.51 to 0.91), with moderate values for maximum angular jerk (ICC = 0.51). For the separate repetitions, ICC values for maximum angular jerk (ICC = 0.30) and sample entropy of acceleration (ICC = 0.28) and angular velocity (ICC = 0.41) were poor. For the forearm sensor, the results were similar to the hand sensor. ICC values were high apart from maximal angular jerk, which was moderate (ICC = 0.69). For the separate repetitions, ICC values for were high apart from maximal angular jerk (ICC = 0.65), and sample entropy of acceleration (ICC = 0.32) and angular velocity (ICC = 0.38). For the upper arm sensor, ICC values were moderate to excellent for the full-task signal, with the exception of maximal jerk (ICC = 0.38). For the separate repetitions, maximal jerk (ICC = 0.50) and sample entropy of acceleration (ICC = 0.38) and angular velocity (ICC = 0.33) were poor. 

Within-session reliability: Reach-and-grasp vertical—All full-task signal ICC values for the hand sensor ranged from moderate to excellent. For the separate repetitions, ICC values were lower for maximal jerk (ICC = 0.42) and maximal angular jerk (ICC = 0.25) as well as for sample entropy of acceleration (ICC = 0.29) and angular velocity (ICC = 0.26). All full-task signal ICC values for the forearm sensor ranged from moderate to excellent with the exception of sample entropy of acceleration and angular velocity (ICC = 0.42). For the separate repetitions, ICC values were lower for maximal jerk (ICC = 0.32), maximal angular jerk (ICC = 0.27), maximal acceleration (ICC = 0.40), and sample entropy of acceleration (ICC = 0.18) and angular velocity (ICC = 0.36). For the upper arm sensor, ICC values were moderate to good, with the exception of sample entropy of acceleration (ICC = 0.33) and angular velocity (ICC = 0.42). For the separate repetitions, ICC values were similar in comparison with the full-task signal. ICC values were lower for maximal jerk (ICC = 0.49), maximal angular jerk (ICC = 0.39), and sample entropy of acceleration (ICC = 0.32) and angular velocity (ICC = 0.40).

Within-session reliability: Reach sideways—For the hand sensor, ICC values were higher for the full-task signal in comparison to the separate repetitions. For the full-task signal, all ICC values were good to excellent. For the separate repetitions, ICC values were lower for maximal jerk (ICC = 0.47), maximal angular jerk (ICC = 0.35), and sample entropy of acceleration (ICC = 0.29) and angular velocity (ICC = 0.39). For the forearm sensor, ICC values were higher for the full-task signal in comparison with the separate repetitions. For the full-task signal, all ICC values were moderate to excellent. For the separate repetitions, ICC values were lower for maximal jerk (ICC = 0.48), maximal angular jerk (ICC = 0.37), and sample entropy of acceleration (ICC = 0.47) and angular velocity (ICC = 0.50). For the upper arm sensor, ICC values were moderate to excellent for the full-task signal, with the exception of maximal angular velocity (ICC = 0.42). For the separate repetitions, ICC values were lower for maximal angular jerk (ICC = 0.46) and sample entropy of acceleration (ICC = 0.40). 

Between-session reliability: Reach forwards—For the hand sensor, ICC values from the full-task signal were good, with the exception of maximal angular jerk (ICC = 0.60) and mean acceleration (ICC = 0.68), which were moderate. For the separate repetitions, ICC values for mean acceleration (ICC = 0.65) and sample entropy of acceleration (ICC = 0.66) were moderate, and the value for mean angular jerk was poor (ICC = 0.34), but the remaining values were all good. For the forearm sensor, all ICC values for the full-task signal and separate repetitions were good to excellent. For the upper arm sensor, all full-task signal ICC values were good to excellent. For the separate repetitions, ICC values were good to excellent, except for sample entropy of angular velocity (ICC = 0.15). 

Between-session reliability: Reach-and-grasp vertical—For the hand sensor, ICC values from the full-task signal were moderate to excellent except for sample entropy of acceleration (ICC = 0.35) and angular velocity (ICC = 0.42). For the separate repetitions, ICC values were moderate to excellent, with the exception of sample entropy of angular velocity (ICC = 0.47). For the forearm sensor, maximal jerk (ICC = 0.66), maximal acceleration (ICC = 0.61), and RMS of acceleration (ICC = 0.60) were moderate, whereas the remaining ICCs were good to excellent. For the separate repetitions, ICC values were good to excellent, with the exception of mean angular velocity (ICC = 0.49). For the upper arm sensor, ICC values from the full-task signal were moderate to excellent, with the exception of sample entropy of acceleration (ICC = 0.41). For the separate repetitions, ICC values were poor to moderate, with only mean (ICC = 0.77) and RMS (ICC = 0.80) of angular velocity being good. 

Between-session reliability: Reach sideways—For the hand sensor, ICC values were good to excellent, with the exception of mean (ICC = 0.58) and RMS (ICC = 0.63) of acceleration and mean angular velocity (ICC = 0.67). For the separate repetitions, ICC values were moderate to good, with the exception of sample entropy of acceleration (ICC = 0.26). For the forearm sensor, ICC values were moderate to good for the full-task signal and moderate for the separate repetitions. For the upper arm sensor, ICC values were moderate to excellent for the full-task signal and moderate to good for the separate repetitions. 

### 3.3. Discriminative Validity

Median and inter-quartile ranges of the IMU parameters for the TD and DCP groups, as well as the *p*-values for the group comparison, are presented in Table 1.

The differences between the DCP and TD groups showed a similar pattern for the hand and forearm sensors. Maximal jerk, maximal angular jerk, and maximal acceleration were significantly higher (*p* < 0.001) for the participants with DCP during all tasks. Maximal angular velocity was also significantly higher for the participants with DCP during all tasks (*p* < 0.004). Mean, RMS, and sample entropy of acceleration and angular velocity were not significantly different between groups. 

For the upper arm sensor, maximal jerk, angular jerk, and acceleration were significantly higher (*p* < 0.001) for the participants with DCP during all tasks. Mean acceleration and angular velocity, as well as sample entropy, were not significantly different between groups. RMS of acceleration was significantly higher for participants with DCP during all tasks: (RF: 1.69 vs. 1.31 m/s²; *p* = 0.049); RGV (1.76 vs. 1.30 m/s²; *p* = 0.009); RS (1.89 vs. 1.54 m/s²; *p* = 0.008). Maximal angular velocity was significantly higher for participants with DCP during RGV (4.20 vs 3.11 rad/s; *p* < 0.001) and RS (4.89 vs. 4.00 rad/s; *p* = 0.034). 

The mean absolute difference between the TD and DCP groups and the between-session SEM are presented in Figure 4 and Appendix A. For sample entropy, the between-session SEM was higher than the between-group difference for all sensors and all tasks. For maximal jerk, the between-group difference was higher than the between-session SEM for all sensors and tasks, but this was not the case for maximal angular jerk. For mean acceleration, the results were task-dependent. During RF, the between-session SEM was higher than the between-group difference for all sensors. During RGV and RS, the between-group difference exceeded the between-session SEM for the hand and upper arm sensors but not for the forearm sensor. For maximal acceleration, the between-group difference was higher than the between-session SEM for all sensors and all tasks. This was also the case for RMS of acceleration, albeit with smaller differences than for maximal acceleration.

## 4. Discussion

This study aimed to assess whether IMUs can reliably measure movement characteristics and whether they can efficiently discriminate between individuals with and without DCP. We found that most of the selected parameters were reliable within and between sessions and can discriminate the movement patterns from individuals with DCP from their TD peers.

The first aim of this study was to evaluate within- and between-session reliability of sensor parameters in individuals with DCP. IMUs have often been used for the calculation of spatio-temporal parameters during gait, such as stride time, stride length, gait events, and gait velocity, in both pathological and non-pathological populations [37,38,39]. However, very few studies have explored the reliability of IMU measures during functional upper limb tasks. Studies focusing on upper limb movements have mostly evaluated the reliability of range of motion [40,41] or 3D trajectories [42] in both healthy and pathological populations. The current study did not focus on spatial measures such as joint angles because the sensor fusion algorithms required to obtain correct joint angles from acceleration and angular velocity values are complicated for upper limb joint angles and have not been validated in pathological populations such as CP. To our knowledge, only Lakerveld et al. have explored the reliability of sensor metrics during upper limb tasks in healthy adults, reporting ICC values of between 0.29 and 0.90 for different functional tasks [43].

Overall, the reliability of the sensor parameters for the full-length signal was good to excellent for all tasks for the hand sensor, and moderate to excellent for the forearm and upper arm sensors. During RGV, we found lower ICC values for sample entropy for the forearm and upper arm sensors, which could be attributed to the increased difficulty of the RGV task, triggering more deviant movement patterns with more variability in signal complexity. There was no distinguishing pattern when comparing tasks with one another, i.e., the ICC values were similar for all three tasks. For the separate repetitions, ICC values were overall slightly lower, with the largest discrepancies in sample entropy and maximal jerk for all tasks. Raw data inspection confirmed that the values of sample entropy showed low variability between trials, impacting the ICC values in a detrimental manner [34]. Additionally, the discrepancy in ICC values for short and long signals proves that sample entropy and its consistency is dependent on signal length and should therefore not be used in short signals.

The between-session reliability ICCs were overall moderate to excellent, with the main outliers being sample entropy for the upper arm sensor during RF and RGV. Caution is thus warranted when interpreting this parameter, keeping in mind the lack of variability in scores, which is a prerequisite for the use of ICC values. In general, the moderate to good ICC values for within- and between-session reliability show that the selected parameters can be reliably measured over time, whilst warranting caution with the interpretation of sample entropy values, as this parameter was less reliable within one session for a short signal, and between sessions for the upper arm sensor.

The second aim of this study was to evaluate whether sensor parameters are sufficiently sensitive to discriminate between the upper limb movement patterns of participants with DCP and their TD peers. Participants with DCP showed higher maximal jerk and angular jerk than their TD peers, which can be attributed to the more jerky movements we expect for involuntary movements as defined in DCP. These results are in agreement with Sanger et al., who found a higher jerk index for participants with DCP during outward reaching [11]. It is noteworthy that jerk has previously often been derived from position measures, which contains more noise than the jerk measure derived from linear acceleration, as this reduces the number of derivations [11]. Jerk measures are therefore promising parameters to differentiate participants with DCP from their TD peers. However, future research is necessary to assess how individuals with DCP deviate from, e.g., individuals with spastic CP. The higher maximal acceleration and angular velocity for the DCP group can additionally stem from the consistent presence of involuntary movements in the extremities. These parameters have previously only been included in studies with PD patients where mainly lower acceleration and angular velocity was found [44]. The comparison of the between-session measurement error and the mean absolute difference between the TD and DCP groups confirmed the discriminative ability of maximal jerk, maximal acceleration, and angular velocity during all tasks, and of the RMS of acceleration during RS. These parameters could in the future be used to detect more specific differences between movement disorders, e.g., between dystonia and choreoathetosis. Overall, there were no task-dependent differences, indicating that the sensitivity of the sensor parameters was retained for the different functional tasks and that these parameters could be exported to different (e.g., home-based) functional tasks in future research.

While valuable insights on the use of sensors to evaluate functional upper limb tasks in individuals with DCP emerged, this study warrants some critical reflections. First, the age range of the included participants was 5–25 years, which is wide. However, this age range was chosen to include a variety of participants with differing levels of dystonia and choreoathetosis severity and varying MACS levels, considering that DCP is a rather rare condition and that individuals with DCP who can perform functional upper limb movements are a minority [5]. However, this also implies that we included varying levels of maturity in reaching and grasping patterns, as reported by Simon-Martinez et al. [45], who showed that upper limb reaching patterns in TD children varied between 5–7 y olds and 8–10 y olds, to finally evaluate into a mature reaching and grasping pattern from the age of 12. Second, due to the experimental nature of this study, data collection was performed in a laboratory setting. Thus, it is possible that participants may have been less comfortable in comparison to their home and/or school environment. Future studies should therefore explore the use of IMUs in the participants’ natural environment, and increase the focus on long-term measurement. This combination would ensure the optimal replication of daily-life fluctuations in rest and action, aided by the fact that dystonia and choreoathetosis are exacerbated by increased stress and emotional arousal [1].

## 5. Conclusions

This study was the first to assess the reliability of sensor parameters in individuals with DCP, reporting high between- and within-session reliability for the majority of the sensor parameters. These findings suggest that the pathological movements of individuals with DCP can be reliably captured using a selection of sensor parameters. The between-group differences highlighted that a selection of sensor parameters is able to discriminate pathological from non-pathological movement patterns, which creates opportunities for further exploration of the pathological movement patterns in individuals with DCP. 

This work contributes to the first steps towards an increased understanding of the complex pathological movement patterns in participants with DCP, while simultaneously stepping away from expensive, location-restricted movement laboratories.

## Figures and Tables

**Figure 1 sensors-23-01574-f001:**
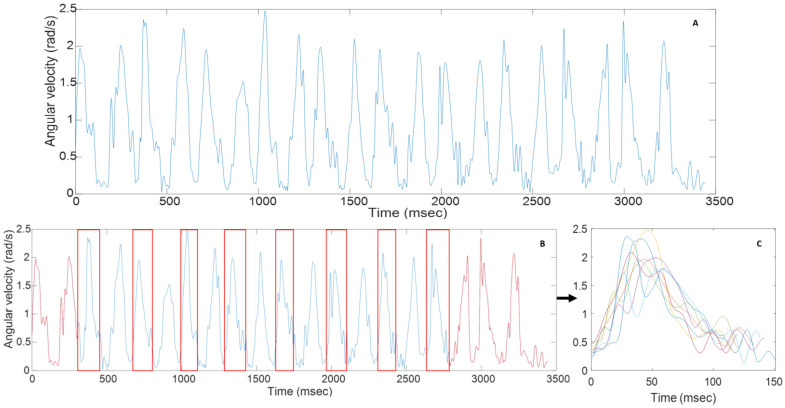
Full-length angular velocity signal (**A**), timings of separate repetitions with the first and last repetition in red (**B**), and an example of the eight segmented repetitions (red boxes) of a reach-forward task (**C**).

**Figure 2 sensors-23-01574-f002:**
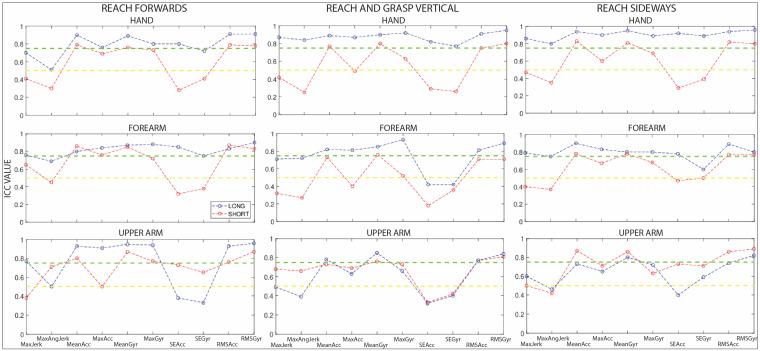
Within-session ICC values of the different sensor parameters for reach forwards, reach-and-grasp vertical, and reach sideways. The green and yellow dotted lines depict the limit of good and moderate correlations, respectively. Blue and red lines are for illustrative purposes. Acc = acceleration; Gyr = angular velocity; Max = maximal; RMS = root mean square; SE = sample entropy.

**Figure 3 sensors-23-01574-f003:**
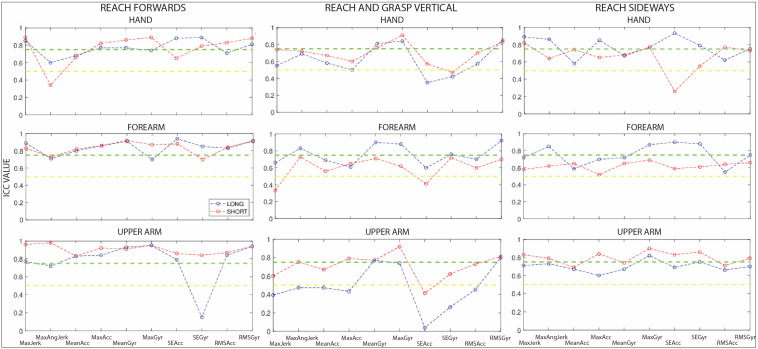
Between-session ICC values of the different sensor parameters for reach forwards, reach-and-grasp vertical, and reach sideways for the DCP group. The green and yellow dotted lines depict the limit of good and moderate correlations, respectively. Blue and red lines are for illustrative purposes.

**Figure 4 sensors-23-01574-f004:**
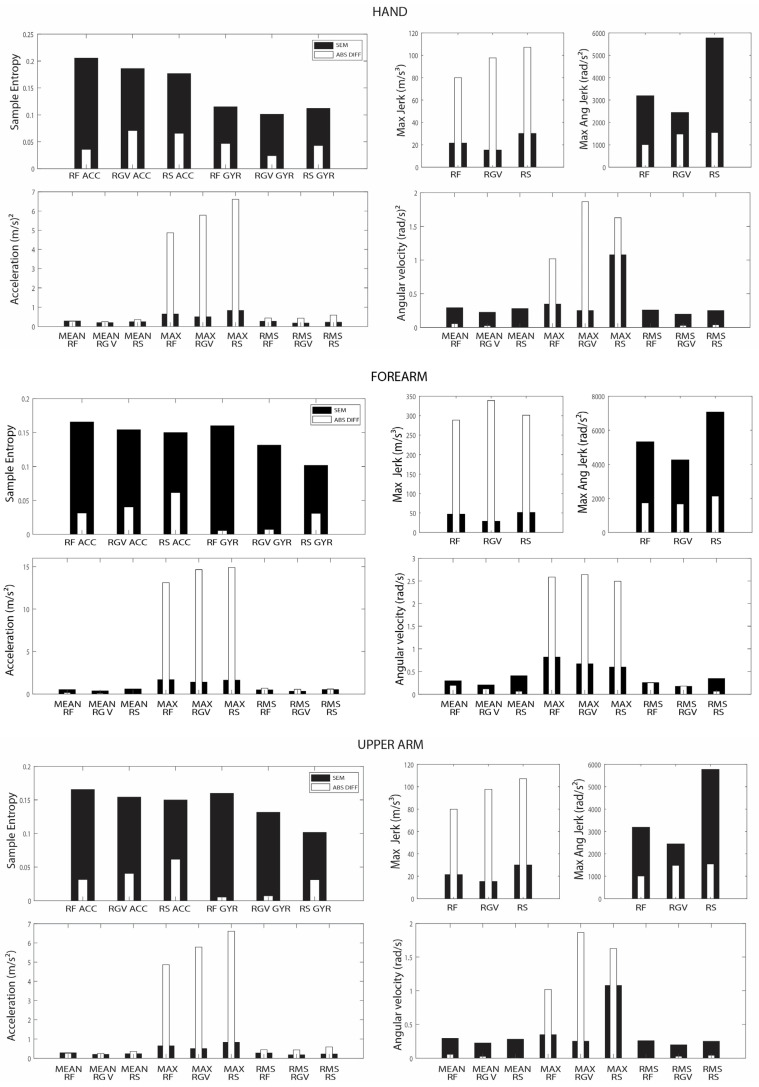
Between-session standard error of measurement (SEM) and absolute mean differences between the TD and DCP groups for forearm, hand, and upper arm sensors. RF = reach forward; RGV = reach-and-grasp vertical; RS = reach sideways; MAX = maximal; RMS = root mean square. SEM = standard error of measurement. ABS DIFF = absolute difference.

**Table 1 sensors-23-01574-t001:** Median and interquartile ranges for sensor parameters as well as *p*-values for between-group differences. TD = typically developing; DCP = dyskinetic cerebral palsy; Acc = acceleration; Gyr = gyroscope; Max = maximal; RMS = root mean square; SE = sample entropy. Significant *p*-values (<0.05) in bold.

	Hand			Forearm			Upper Arm		
	TD	DCP	*p*-Value	TD	DCP	*p*-Value	TD	DCP	*p*-Value
**Reach Forward**									
Max Jerk (m/s³)	259.60 (66.84)	713.39 (428.07)	**<0.001**	173.18 (36.68)	474.98 (334.60)	**<0.001**	64.18 (18.70)	117.48 (74.70)	**<0.001**
Max Ang Jerk (rad/s²)	2132.77 (933.58)	4367.23 (1992.18)	**<0.001**	1347.84 (338.62)	2771.37 (1468.10)	**<0.001**	896.54 (268.62)	1562.82 (809.96)	**<0.001**
Mean Acc (m/s²)	2.52 (0.67)	2.67 (1.36)	0.792	2.37 (0.49)	2.44 (1.09)	0.880	1.16 (0.50)	1.39 (0.68)	0.096
Max Acc (m/s²)	13.30 (4.89)	31.58 (17.07)	**<0.001**	9.93 (2.35)	23.76 (13.13)	**<0.001**	4.56 (1.70)	7.61 (3.95)	**<0.001**
Mean Gyr (rad/s)	1.34 (0.39)	1.50 (0.74)	0.235	1.03 (0.30)	1.14 (0.62)	0.513	1.02 (0.31)	0.95 (0.54)	0.821
Max Gyr (rad/s)	4.65 (1.81)	8.41 (3.50)	**<0.001**	3.97 (1.22)	5.76 (2.48)	**<0.001**	3.43 (1.44)	3.91 (1.35)	0.144
RMS Gyr (rad/s)	1.71 (0.32)	2.14 (1.03)	0.101	1.36 (0.41)	1.61 (0.90)	0.392	1.43 (0.43)	1.26 (0.67)	0.860
RMS Acc (m/s²)	3.41 (0.60)	4.26 (2.19)	0.089	2.93 (0.46)	3.49 (1.60)	0.118	1.31 (0.59)	1.69 (0.88)	**0.049**
SE Acc	0.15 (0.22)	0.15 (0.10)	0.588	0.18 (0.21)	0.18 (0.14)	0.481	0.29 (0.31)	0.26 (0.20)	0.465
SE Gyr	0.14 (0.20)	0.16 (0.11)	0.989	0.13 (0.13)	0.17 (0.09)	0.546	0.12 (0.08)	0.16 (0.11)	0.066
**Reach and Grasp Vertical**									
Max Jerk (m/s³)	265.79 (125.95)	713.80 (471.34)	**<0.001**	178.05 (58.59)	510.68 (365.21)	**<0.001**	61.49 (18.62)	139.02 (85.14)	**<0.001**
Max Ang Jerk (rad/s²)	2303.27 (825.93)	4650.33 (2397.23)	**<0.001**	1684.83 (652.87)	2561.37 (980.48)	**<0.001**	865.37 (272.50)	1778.84 (1007.55)	**<0.001**
Mean Acc (m/s²)	2.26 (0.65)	2.15 (0.82)	1.000	2.24 (0.57)	2.15 (0.91)	0.958	1.06 (0.56)	1.42 (0.41)	0.054
Max Acc (m/s²)	13.62 (6.23)	32.80 (17.02)	**<0.001**	10.76 (3.34)	23.77 (14.45)	**<0.001**	4.15 (1.01)	8.49 (3.50)	**<0.001**
Mean Gyr (rad/s)	1.33 (0.41)	1.35 (0.52)	1.000	0.93 (0.24)	0.92 (0.70)	0.979	0.92 (0.28)	0.89 (0.37)	1.000
Max Gyr (rad/s)	6.26 (1.36)	9.56 (3.22)	**<0.001**	4.51 (0.82)	6.26 (3.25)	**0.004**	3.11 (0.95)	4.20 (1.96)	**<0.001**
RMS Gyr (rad/s)	1.82 (0.33)	2.05 (0.65)	0.290	1.34 (0.37)	1.41 (0.66)	0.657	1.24 (0.40)	1.24 (0.54)	0.855
RMS Acc (m/s²)	2.98 (0.68)	3.75 (1.98)	0.067	2.71 (0.45)	3.11 (1.99)	0.118	1.30 (0.58)	1.76 (0.66)	**0.009**
SE Acc	0.18 (0.19)	0.14 (0.11)	0.133	0.25 (0.19)	0.16 (0.15)	0.192	0.37 (0.24)	0.28 (0.23)	0.106
SE Gyr	0.15 (0.17)	0.12 (0.10)	0.443	0.14 (0.14)	0.15 (0.13)	0.917	0.13 (0.06)	0.14 (0.10)	0.498
**Reach Sideways**									
Max Jerk (m/s³)	233.92 (58.71)	590.64 (299.39)	**<0.001**	185.14 (84.70)	442.08 (313.69)	**<0.001**	64.58 (27.32)	139.74 (113.24)	**<0.001**
Max Ang Jerk (rad/s²)	2572.43 (1200.00)	4926.06 (3111.24)	**<0.001**	1403.92 (613.41)	2913.19 (2164.19)	**<0.001**	1058.38 (320.31)	2172.85 (1033.44)	**<0.001**
Mean Acc (m/s²)	3.13 (0.96)	3.00 (1.52)	0.627	2.88 (0.71)	2.92 (1.29)	0.743	1.28 (0.34)	1.57 (0.69)	**0.022**
Max Acc (m/s²)	12.35 (4.09)	27.73 (13.35)	**<0.001**	9.82 (3.56)	22.44 (16.34)	**<0.001**	4.38 (0.97)	8.90 (4.93)	**<0.001**
Mean Gyr (rad/s)	1.83 (0.66)	1.96 (0.58)	0.728	1.54 (0.59)	1.63 (0.56)	0.960	1.13 (0.41)	1.12 (0.42)	0.860
Max Gyr (rad/s)	5.72 (1.47)	9.03 (4.59)	**<0.001**	5.04 (0.90)	7.18 (3.63)	**0.002**	4.00 (0.63)	4.89 (1.56)	**0.034**
RMS Gyr (rad/s)	2.23 (0.55)	2.53 (0.98)	0.461	1.98 (0.65)	2.15 (0.77)	0.940	1.51 (0.53)	1.46 (0.58)	0.513
RMS Acc (m/s²)	3.85 (0.93)	4.65 (1.79)	0.336	3.34 (0.86)	3.85 (2.12)	0.339	1.45 (0.38)	1.89 (0.78)	**0.007**
SE Acc	0.18 (0.26)	0.17 (0.12)	0.380	0.23 (0.27)	0.22 (0.13)	0.268	0.32 (0.32)	0.27 (0.17)	0.227
SE Gyr	0.16 (0.18)	0.17 (0.13)	0.647	0.13 (0.13)	0.14 (0.13)	0.208	0.12 (0.09)	0.14 (0.07)	0.190

## Data Availability

The data presented in this study are available in the Appendix A.

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
