# Peer review of "Reliability and Discriminative Validity of Wearable Sensors for the Quantification of Upper Limb Movement Disorders in Individuals with Dyskinetic Cerebral Palsy"

_sensors, 2023, doi:10.3390/s23031574_

Round 1

Reviewer 1 Report

The submitted manuscript is about the reliability and discriminative validity of measuring upper limb movement disorders by use of IMUs. Assessing the measurement qualities for use in health care is an important research field for enabling the transfer to clinical practice. Some aspects could be addressed in a revision, as listed in the following:

- abstract: the conclusions seem to be vague and not directly related to the study results. 

- introduction: for understanding the purpose of measuring arm motions, it would be helpful to learn more, which specific movement phenomena are expected in the investigated population including more details on how these phenomena are assessed in usual care.

- what is the MACS, what is being tested, how and what is known concerning the clinimetric properties of this assessment?

- what was the purpose of considering different movement tasks? There is no research question outlined addressing this aspect.

- methods: there seems to be a focus on smoothness and speed-based metrics and no consideration of spatial measures. What was the reasoning of the authors for this selection?

- please state the specific system of the company being used 

- figure 1a shows 4 IMUs placed on the upper arm, forearm, hand and sternum, but only three sensors are described in the manuscript

- what was the idea that motivated the investigation of only the upper limb and forearm sensor?

- how was the sensor data extracted (was raw accelerometry being used)?

- could the authors give some indication of when to consider a metric reliable, based on ICCs?

- what was the basis for the metric selection. Were all investigated intended to measure similar movement aspects or different features?

- results and discussion: what was the intention for the separate descriptions for the upper and lower arm sensor?

- how do the authors interpret the poor results on some metric ICCs? Are any conclusions to be drawn for usage or non-use in clinical practice?

Reviewer 2 Report

Please describe the parameters that IMUs (XSens, the Nether- 126 lands) measures. What is the accuracy of the device?

There is no information regarding the video system. What is the purpose of this system in the context of measurements?  

What is the influence of sensor placement in measured angular parameters of the upper arm?

Please refer to the rotational movement parameters (the equations 1 and 2 but not only) using the standard symbolization of angle, angular velocity, angular acceleration and jerk. (Theta, omega, alpha and zeta)

From biomechanical perspective, the axis of rotation and the linear axis of the IMUs are associated with which directions, segments, joints? On one instance you consider the “upper arm and hand” which are body segments, while the third one is “wrist”? A body joint? Please clarify. Also, the units of the parameters in the tables 1 and A3  are not consistent “Max Ang Jerk (rad/s²)”
